# Direct RNA sequencing of respiratory syncytial virus infected human cells generates a detailed overview of RSV polycistronic mRNA and transcript abundance

I'ah Donovan-Banfield[1,2], Rachel Milligan[1], Sophie Hall[1], Tianyi Gao[1], Eleanor Murphy[1], Jack Li[1], Ghada T. Shawli[2], Julian Hiscox[2], Xiaodong Zhuang[3], Jane A. McKeating[3,4], Rachel Fearns[5]*, David A. Matthews[1]*

1 School of Cellular and Molecular Medicine, Faculty of Life Sciences, University of Bristol, Bristol, United Kingdom, 2 Department of Infection Biology and Microbiome, Institute of Infection, Veterinary and Ecological Sciences, University of Liverpool, Liverpool, United Kingdom, 3 Nuffield Department of Medicine, University of Oxford, Oxford, United Kingdom, 4 Chinese Academy of Medical Sciences Oxford Institute, University of Oxford, Oxford, United Kingdom, 5 Department of Microbiology, National Emerging Infectious Diseases Laboratories, Boston University School of Medicine, Boston, Massachusetts, United States of America

* d.a.matthews@bristol.ac.uk (DAM); rfearns@bu.edu (RF)

**Data Availability Statement:** Raw fastq files (MRC-5 cells data), associated fasta files and poly A length output files are available from Zenodo, DOI: https://doi.org/10.5281/zenodo.5799655. The

## Abstract

To characterize species of viral mRNA transcripts generated during respiratory syncytial virus (RSV) infection, human fibroblast-like MRC-5 lung cells were infected with subgroup A RSV for 6, 16 and 24 hours. In addition, we characterised the viral transcriptome in infected Calu-3 lung epithelial cells at 48 hours post infection. Total RNA was harvested and polyadenylated mRNA was enriched and sequenced by direct RNA sequencing using an Oxford nanopore device. This platform yielded over 450,000 direct mRNA transcript reads which were mapped to the viral genome and analysed to determine the relative mRNA levels of viral genes using our in-house ORF-centric pipeline. We examined the frequency of polycistronic readthrough mRNAs were generated and assessed the length of the polyadenylated tails for each group of transcripts. We show a general but non-linear decline in gene transcript abundance across the viral genome, as predicted by the model of RSV gene transcription. However, the decline in transcript abundance is not uniform. The polyadenylate tails generated by the viral polymerase are similar in length to those generated by the host polyadenylation machinery and broadly declined in length for most transcripts as the infection progressed. Finally, we observed that the steady state abundance of transcripts with very short polyadenylate tails less than 20 nucleotides is less for N, SH and G transcripts in both cell lines compared to NS1, NS2, P, M, F and M2 which may reflect differences in mRNA stability and/or translation rates within and between the cell lines.

## Introduction

Respiratory syncytial virus (RSV) causes a respiratory infection that leads to significant levels of morbidities and mortalities in infants and young children across the globe [1, 2]. Recovery

raw fastq files and poly A length data from Calu-3 cells can be found at Zenodo, DOI: https://doi.org/10.5281/zenodo.7101886.

**Funding:** DAM received award BB/M02542X/1 from the Biotechnology and Biological Sciences Research Council. JAM received funding from the Wellcome Trust (200838/Z/16/Z) and the Medical Research Council (MR/R022011/1). The funders had no role in study design, data collection and analysis, decision to publish, or preparation of the manuscript.

**Competing interests:** The authors have declared that no competing interests exist.

from infection does not lead to long term protection and repeat reinfections over an individual's lifetime are a hallmark of RSV [3, 4]. Thus, hospital admission of elderly patients with life-threatening complications of RSV infection are also common [5, 6]. Typically, RSV represents a global viral respiratory disease burden comparable to that of influenza, highlighting the importance of understanding this virus' lifecycle [7–9].

The RSV genome is single stranded, negative sense RNA approximately 15,000 nt in length. The genome codes for 10 major capped and polyadenylated mRNAs in the order *NS1*, *NS2*, *N*, *P*, *M*, *SH*, *G*, *F*, *M2* and *L* (Fig 1A) [10–12]. The 10 genes code for corresponding proteins except M2 which codes for both M2-1 (the 5′ most open reading frame) and M2-2 which is expressed by an unusual ribosome shunting mechanism and does not have a separate transcript [13]. In virus particles and infected cells, the viral genome exists as a nucleocapsid structure in which the RNA is coated by the nucleocapsid protein (N), and associated with a viral RNA dependent RNA polymerase known as L, a phosphoprotein (P) and viral protein M2-1 [14, 15]. Transcription of the RSV genome depends on the polymerase complex recognizing and responding to *cis*-acting elements within the viral genome. Each of the genes is flanked with conserved elements referred to as gene start (GS) and gene end (GE) signals [12]. Most of the genes are separated by short intergenic regions, although there is one overlapping gene junction, in which the GS signal of the downstream L gene lies 48 nt upstream of the GE signal of the preceding M2 gene [16].

The 3′ end of the genome contains a promoter referred to as the leader or Le promoter. Studies examining the effect of UV exposure on RSV gene expression showed that genes proximal to the 3′ end of the genome were more resistant to UV damage than genes located in the 5′ end of the genome [17]. This finding established that RSV genes are transcribed sequentially from the 3′ to the 5′ end of the genome. Subsequent minigenome studies confirmed that transcription of a downstream gene is dependent on transcription of an upstream gene, and that none of the intergenic regions contains a promoter [18–20]. Based on these and other studies, the prevailing model for RSV transcription is that the polymerase initiates each cycle of transcription at the Le promoter. Subsequent studies showed that the polymerase initiates transcription opposite position 3C of the Le promoter that generates and releases a short, heterogeneous RNA transcript of ~ 20–25 nt in length. The polymerase then scans to locate

A

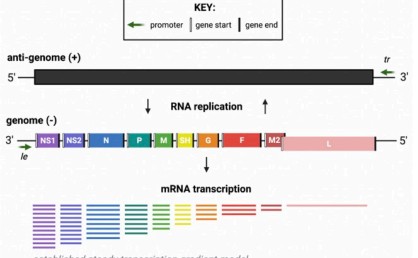

B

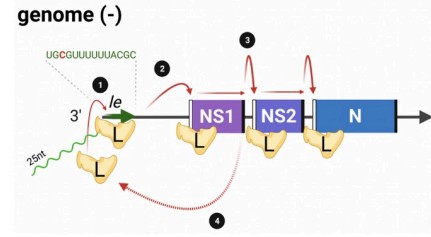

**Fig 1. RSV genome and mRNA transcription.** (A) A schematic of RSV genome organisation, replication and the mRNA transcripts generated by a simple gradient of transcript initiation. *Le*, leader; *tr*, trailer. (B) A zoomed in simplified schematic of the viral RdRp, L, binding to the negative-sense genome to transcribe viral mRNA. L binds to the *le* promoter [1] and initiates transcription opposite position 3 of the Le promoter, generating a ~20–25 nt transcript (shown in green) which is released before the polymerase scans for the *NS1* gene start signal (vertical white line) [2] from here the mRNA is transcribed until it reaches a gene end signal (vertical black line) and poly-adenylates the nascent mRNA. L will then either continue scanning along to the next gene start signal [3] or fall off and have to re-associate with the genome at the le promoter and start RNA synthesis again [4].

the GS signal of the first gene where it reinitiates RNA synthesis and caps the mRNA. The polymerase then elongates the mRNA until it reaches a GE signal where it polyadenylates the mRNA by a reiterative stuttering mechanism, and then releases it (Fig 1B) [21, 22]. Occasionally, the polymerase fails to recognize a GE signal, resulting in the synthesis of a polycistronic mRNA. The possibility also exists that the polymerase could fail to re-initiate RNA synthesis at a GS signal. It is thought that in this case, the polymerase dissociates from the viral genome ending that cycle of mRNA transcription. Because initiation of a new cycle of transcription depends on the polymerase associating with the Le promoter, a transcription gradient is established with genes that are proximal to the 3′ end of the genome molecule being transcribed more than those nearer the 5′ end.

Evidence for a transcription gradient was originally obtained by quantifying mRNAs generated in an *in vitro* transcription system and from studies using a small molecule polymerase inhibitor that prevented initiation at the promoter [23, 24]. However, there have been several attempts to quantify the relative expression of RSV mRNAs which do not fully agree with these studies. Several reports used Illumina based RNA-seq analysis of intracellular viral RNA transcripts, which showed a deviation from the expected transcription gradient [25]. In another study, RT-qPCR analysis of mRNA levels in RSV infected cells revealed genotype-specific variations in mRNA levels, but with *G* having a higher mRNA level than *N* in each genotype tested, a finding that could not be accounted for by differential RNA stability [26]. Based on these findings, the authors proposed that the polymerase could scan through upstream genes in a non-transcribing mode and then initiate at a downstream GS signal, and/or that some gene junction signals facilitate polymerase recycling so that certain genes are transcribed reiteratively [26]. In this model, after reaching the GE signal for the G gene, for example, the polymerase might scan upstream to re-initiate mRNA transcription at the G gene GS signal, this would entail almost 1000 nt of upstream scanning. In support of this it has been shown that the polymerase can scan upstream from the M2 GE signal to locate the GS signal for the L gene [19] but this is a relatively small distance of less than 68 nt (in the case of RSV-A). However, these recent studies suffer to one degree or another from confounding factors. For example, reverse transcription and PCR amplification, which are necessary steps in Illumina based RNA-seq experiments, could be variable in efficiency depending on the mRNA transcript sequence and secondary structure. Likewise, RT-qPCR based quantification of different viral genes could be influenced by the selection of standards, used to determine copy number, that do not accurately mimic viral mRNAs.

Direct RNA sequencing (dRNAseq) has been used to study transcriptomes from a range of sources including human and viral infections [27–32]. The technique sequences mRNA directly from the polyadenylated tail towards the 5′ cap by feeding the mRNA molecule through a nanopore and measuring changes in current across the pore as nucleotides pass through. Inherently dRNAseq can suffer from a 3′ bias as it is not possible to determine if the molecule sequenced is full length with an authentic cap or if it is degraded. In addition, there is a significant error rate where approximately 10% of nucleotides are wrong or missing. Nonetheless, the technique offers a direct measurement of mRNA abundance directly without the potentially confounding experimental steps (e.g. reverse transcription and PCR) and allows sequencing of the entire mRNA molecule that provides important imformation on its structure, whether it is polycistronic and poly A tail length [33].

We have previously used dRNAseq to examine the transcriptome of human adenoviruses, SARS-CoV-2 and adenovirus vector-based vaccines illustrating the significant advantages this approach offers in a range of different settings [27, 31, 34]. Here we apply the technique to examine the transcriptome of human RSV infected of MRC-5 cells sampled over time. We show that there is a gradient of transcription as proposed in early studies but this is not strictly

linear. We also provide insights into the rate of production of viral polycistronic mRNA and show that the poly A tails of RSV mRNA are similar in length to those of human mRNA. To broaden the applicability of our findings we examined the steady state abundance of RSV transcripts in Calu-3 lung epithelial cells sampled at a single time point and show a similar non-linear decline in transcript abundance. These findings cement the utility of dRNAseq for analysing viral transcript frequencies.

## Materials and methods

### Virus and cells

Human MRC-5 cells (a genetically normal human lung fibroblast-like line) Calu-3 (a human lung adenocarcinoma cell line) and HEp2 (a derivative of HeLa cells) cells were obtained from European Collection of Authenticated Cell Cultures (#05072101, ECACC). The cells were cultured in DMEM supplemented with 10% foetal bovine serum, 100 U/ml penicillin and 100 µg/ml streptomycin. Stocks of RSV strain A2 were generated by low multiplicity of infection (0.01 $TCID_{50}$/cell) of Hep2 cells and titrated by $TCID_{50}$. To generate infected and control cell mRNA for this analysis, MRC-5 cells were infected at the University of Bristol with RSV strain A2 at a multiplicity of 3 $TCID_{50}$/cell to ensure high rates of infection across the cell sheet. The infected cells were harvested at 6, 16 and 24 hours post infection (hpi) before extensive cell death is observed. Independently, a second stock of RSV, strain A2 was similarly generated and assayed at the University of Oxford site and subsequently used to infect Calu-3 cells for 48 hours at a multiplicity of 0.5 $TCID_{50}$/cell again before extensive cell death occurs in this cell line.

### RNA extraction and sequencing

RNA extraction and sequencing was as described previously [27, 31, 34]. Briefly, total RNA was extracted from the infected cell using TRIzol reagent (#15596026, Ambion) at 1ml of reagent per $10^7$ cells, as per manufacturer's recommendations except that the final wash of the RNA pellet in 70% ethanol was repeated a further two times (total of 3 washes). Total RNA was enriched for poly A tails using Dynabeads™ mRNA purification kit (#61006, Invitrogen) as per manufacturer's instructions and we used 350ng of poly A enriched RNA per sequencing reaction. We used the SQK-RNA002 kits and MIN106D R9 version flow cells (Oxford Nanopore Technologies), following the manufacturer's protocols.

### Data analysis, mapping to viral genome and human transcripts

Reads were mapped to a fasta file containing the RSV A2 genome (accession number KT992094.1) using minipmap2 [35] (command line example: minimap2 -a -x map-ont -uf -k14—sam-hit-only RSV_AB.fasta RSV2020_16hpi.fastq > map_2_RSV.sam).

### Data analysis, characterisation of viral transcripts

To analyse nanopore transcripts that have a very high error rate we developed a software pipeline that assigns transcripts according to the ORFS they code [27]. Briefly, the pipeline takes the mapping information from the SAM file to group the transcripts mapping to the viral genome according to shared start locations, exons (or equivalent) and stop locations. Once grouped in this manner, a representative transcript is then generated for each transcript group based solely on the genome sequence. User supplied information on the location of known ORFs is employed to assign transcript groups according to what known ORFS would be coded by each transcript group. In this way the pipeline determines the percentage of mRNAs that

express each ORF relative to the total number of transcripts that map to the target genome. This pipeline generates tables describing the structure of transcripts, what ORFs are present and how many mRNA molecules belong to each group of transcripts. Allied to this analysis, nanopolish [33] was used to determine the poly A length of each sequenced transcript and in house scripts grouped transcripts according to GE usage before compiling a list of poly A lengths for generating violin plots. For the ORF centric pipeline analysis, only transcripts that were QC-flagged as PASS by nanopolish and had an estimated poly A tail length of 20 or more were considered (approximately 60% of transcripts). For the analysis involving short poly A lengths, again only transcripts QC-flagged as PASS by nanopolish were considered. Visualisation of differences in poly A tail lengths of transcripts sharing the same GE signal was conducted in R 4.1.2 using RStudio, using ggplot2 (v3.3.5) and tidyverse (v1.3.1) packages.

### Data analysis of G glycoprotein transcripts

Previously, and in line with other groups using dRNAseq, we had noted that the 5′ most 10 or so nucleotides of an mRNA molecule are lost during nanopore sequencing [27, 28]. To compensate for this in our analysis pipeline we use the genome sequence to programmatically add back the missing 10 nucleotides upstream and this approach works well for transcripts where the authentic initiating methionine is close to the 5′ cap. In the case of the G protein of RSV however this approach leads the software to utilise an out of frame AUG some 8 nt upstream of the authentic AUG. The software pipeline thus reports a large number of transcripts in which the apparent primary ORF is short and unknown but is immediately followed by a second ORF which is correctly identified as the G protein. For the purposes of this analysis, we have counted these as genuine G gene transcripts since manual inspection of the structure of these transcripts is consistent with the GS for G protein mRNA.

### Data availability

Raw fastq files (MRC-5 cells data), associated fasta files and poly A length output files are available from Zenodo, DOI: **https://doi.org/10.5281/zenodo.5799655**.

The raw fastq files and poly A length data from Calu-3 cells can be found at Zenodo, DOI: **https://doi.org/10.5281/zenodo.7101886**.

### Code availability

The software for the ORF centric pipeline and for segregating the transcripts by poly A site usage and by poly A site length is available from the authors on request or from Zenodo, DOI: https://doi.org/10.5281/zenodo.7101768. Instructions for running this pipeline are included alongside the software at Zenodo but also available from the authors on request.

## Results

### Mapping RSV transcripts in MRC-5 cells

To examine RSV transcript abundance over time using dRNAseq, MRC-5 cells were infected with RSV strain A2 at an moi of 3 $TCID_{50}$/ cell. Cells were then harvested at 6, 16 and 24 hpi to encompass the approximate length of time for a single cycle of virus replication. RNA was isolated and subjected to dRNAseq. Table 1 lists the number of sequence reads from each timepoint and the number mapping to either the RSV genome or the human transcriptome in each sample. The number of reads mapping to the RSV genome climbed dramatically over the course of infection, from just over 3,000 reads at 6 hpi to over 147,000 reads by 24 hpi.

**Table 1. Read counts and mapping information.**

| Sample | FASTQ count (Longest read: Average length) | Mapped to RSV (Longest mapped read) | Mapped to Human Transcripts (Longest mapped read) |
|---|---|---|---|
| **MRC-5 mock** | 2,195,978 (16,860: 1358.5) | 0 | 1,945,497 (16,859) |
| **MRC-5 6hpi** | 1,577,904 (22,167: 1426.4) | 3192 (8,015) | 1,328,890 (22,166) |
| **MRC-5 16hpi** | 2,380,322 (19,372: 1367.1) | 76,702 (9,880) | 1,992,491 (19,371) |
| **MRC-5 24hpi** | 1,585,580 (12,629: 1145.4) | 147,323 (7,196) | 1,125,920 (12,628) |

Table to show total number of reads, longest read and average read length for each dataset alongside mapping data for reads mapping to human transcripts or the RSV genome. For the FASTQ data we also indicate the longest read length and average read length. For the mapped data we also indicate the length of the longest read mapped.

Broadly, the pattern of RSV reads shows a gradient of transcription, with genes that are proximal to the 3′ Le promoter being more highly represented (Fig 2). In addition, since the sequencing technology reads transcripts poly A tail first, there is greater read depth within each gene near the location of the poly A site. The relative levels of mRNA differed at 6 hpi compared to 16 and 24 hpi, with *NS1* and *NS2* transcripts being more highly represented at 6 hpi than at later times. This could be due to asynchronous viral entry, such that in some cells, only the first two genes had been transcribed by 6 hpi. In addition we cannot exclude the possibility that this early timepoint is measuring encapsidated mRNAs in the infecting innocula, rather than *de novo* synthesized transcripts in the infected cells. In contrast, the read patterns at 16 and 24 hpi are similar to each other.

We used our previously described ORF centric analysis pipeline [27] to examine the number of mRNA molecules that code for a known gene, and Table 2 shows the relative contribution of mRNA that code for each RSV gene. This data is a subset of the total mapped reads shown in Fig 2 in that it only includes dRNAseq reads with a poly A tail of over 20 nt. This 20 nt cut-off is used to increase the confidence that the transcripts being counted are genuine

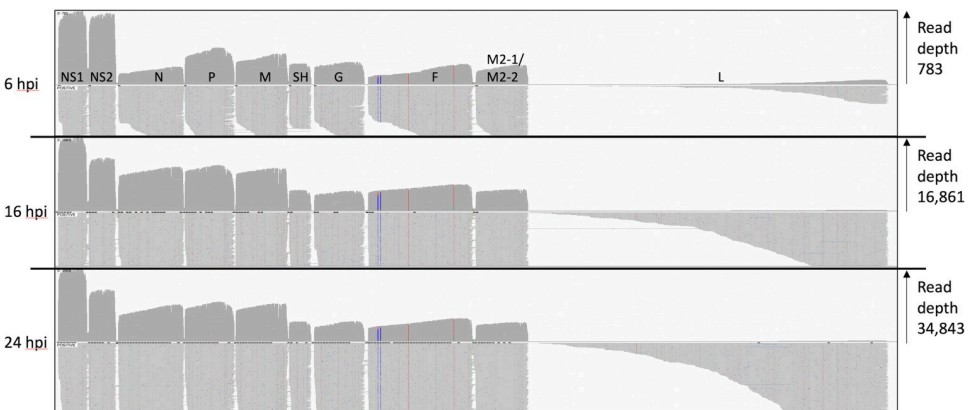

**Fig 2. Reads mapped to the RSV genome.** An IGV generated image of the dRNAseq reads mapped to the viral genome at 6, 16 and 24 hpi, the location of the viral ORFs is indicated above the top panel. Each panel shows the overall depth of reads along the genome in the top half and the bottom half shows a selection of the mapped reads illustrating how they align to the viral genome. The vertical coloured lines within the F ORF pinpoint three SNPs present in the observed reads relative to the reference genome for RSV A2.

**Table 2. List of genes from RSV and their percent contribution to the RSV derived transcriptome in MRC-5 cells.**

| Feature | Gene order on viral genome | 6hpi raw count | 6hpi percent of total (percent of NS1) | 16hpi raw count | 16hpi percent of total (percent of NS1) | 24hpi raw count | 24hpi percent of total (percent of NS1) |
|---|---|---|---|---|---|---|---|
| Not identified | N/A | 563 | 26.61 | 8,639 | 17.26 | 19,587 | 19.82 |
| NS1 | 1 | 406 | 19.19 | 9,078 | 18.13 | 18,926 | 19.16 |
| NS2 | 2 | 323 | 15.26 (80) | 5,123 | 10.23 (56) | 10,539 | 10.67 (56) |
| N | 3 | 75 | 3.54 (18) | 5,308 | 10.60 (58) | 8,604 | 8.71 (45) |
| P | 4 | 201 | 9.50 (50) | 5,537 | 11.06 (61) | 10,111 | 10.23 (53) |
| M | 5 | 139 | 6.57 (34) | 4,823 | 9.63 (53) | 8,630 | 8.73 (45) |
| SH | 6 | 85 | 4.02 (21) | 2,010 | 4.01 (22) | 4,126 | 4.18 (22) |
| G (initiation at 2nd AUG) | 7 | 82 | 3.88 (20) | 1,559 | 3.11 (17) | 3,203 | 3.24 (17) |
| F | 8 | 49 | 2.32 (12) | 2,447 | 4.89 (27) | 3,850 | 3.90 (20) |
| M2 | 9 | 74 | 3.50 (18) | 2,412 | 4.82 (27) | 4,329 | 4.38 (23) |
| L | 10 | 0 | 0 | 5 | 0.01 (0) | 6 | 0.01 (0) |
| Total of all features | | 2,116 | | 50,064 | | 98,801 | |

This table illustrates the number of mapped reads that contain the indicated RSV genes as the 5′ most gene. At each timepoint the raw number of individual transcripts is indicated alongside the percent contribution of those transcripts at that timepoint. In addition, we have calculated the percent abundance of each transcript relative to NS1 abundance. Note that this table shows transcripts with a poly A tail over 20nt long.

mRNA molecules. Notably this table indicates that whilst *NS1* and *NS2* dominate at 6 hpi, only *NS1* seems to dominate thereafter as levels of *NS2* gene expression are then closer to *N* gene expression. Moreover, rather than a gene-by-gene step-wise decline in mRNA abundance the more distal the gene is from the 3′ Le promoter there is instead four groups of gene abundance at 16 and 24 hpi (Table 2 –see abundance relative to NS1 transcripts data). Group 1 containing just NS1, group 2 comprising *NS2*, *N*, *P* and *M* (around half the abundance of NS1) followed by group 3 with *SH*, *G*, *F* and *M2-1/M2-2* (each around 20% of the abundance of NS1) with group 4 containing just L. This matches the visual pattern of depth of aligned transcripts shown in Fig 2 and are consistent with a model of obligatorily sequential transcription. As with our previous publications using dRNAseq [27], there is a large number of transcripts which could not be confidently associated with a particular gene (listed as "not identified") that most likely represent truncated RNA molecules arising from mRNA degradation, mechanical shearing or molecules with short deletions within the body of the transcript. Allied to this observation, whilst we observe patterns of transcript mapping that aligns closely with the expected GS signals, it would be difficult using this approach to identify low level use of any novel GS signals as the technique is insensitive to the 5′ cap structure and thus unable to reliably identify full length transcripts.

## Polycistronic RSV transcripts in infected MRC-5 cells

Previous studies where RSV mRNAs were analysed by Northern blot analysis had revealed the presence of polycistronic species in addition to monocistronic mRNAs [11]. Polycistronic transcripts are generated when the polymerase fails to recognize a GE signal and continues transcribing into the adjacent downstream gene. The dRNAseq data allows a quantification of the abundance of different polycistronic mRNAs. To visually represent polycistronic messages we used an in-house script to group the alignments according to the mapped location of the 3′ end of individual reads (Fig 3)–the nature of dRNAseq means there is greater confidence that the 3′ end of a molecule has been correctly captured. From this data we can clearly see

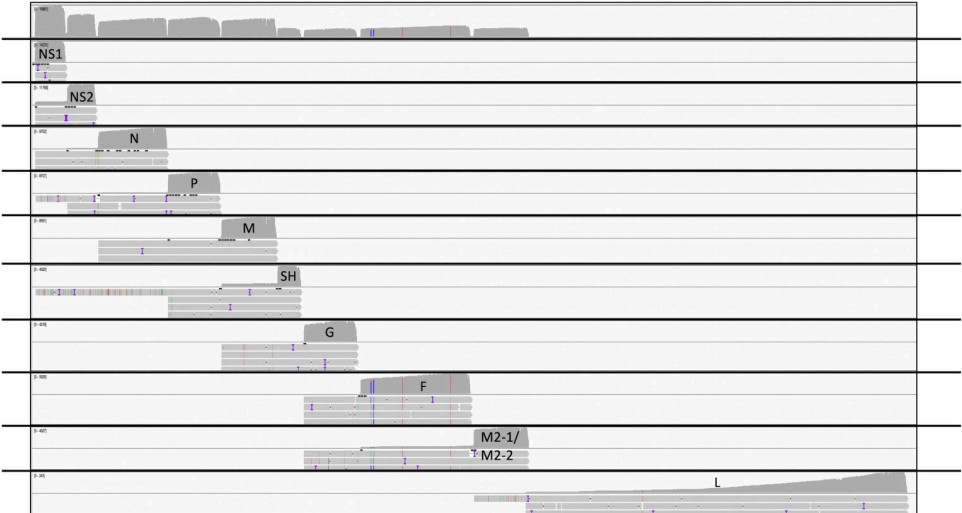

**Fig 3. MRC-5 infected cell reads mapped to the RSV genome grouped by GE site usage.** An IGV generated image of the dRNAseq reads from MRC-5 cells mapped to the viral genome at 24 hpi alongside a series of panels where, in each panel, the reads have been collated according to which GE signal they appear to be utilising. In each panel the top half represents the depth of reads and the bottom half shows a small handful of the longest reads mapping in that group in order to illustrate the presence of polycistronic mRNA in each group.

significant numbers of polycistronic messages, with three or even four genes on the same mRNA molecule in some cases, consistent with previously published Northern blot data. To provide an estimate of the relative rates that a GE signal is ignored to enable the production of a polycistronic message we divided the maximum read depth of the intergenic region by the maximum read depth of the preceding gene (Table 3). The proportion of readthrough at each intergenic region was consistent over time, except for higher levels of readthrough at the *NS1-NS2* and *G-F* intergenic regions at 6 hpi compared to 16 hpi. We note the detection of three broad groups of GE signals, one is characterised by the *NS1*, *M* and *F* GE signals with a relatively high rate of readthrough (9–13%), the second is characterised by the *NS2*, *N*, *P* and *G* GE signals with a more moderate rate of readthrough (2–4%) with the SH gene GE signal being the most effective at signalling mRNA release (only 0.5% readthrough).

Table 4 shows an analysis of polycistronic mRNA detected at 24 hpi in MRC-5 cells where we observe a significant number of polycistronic mRNA produced. This table is calculated

**Table 3. Readthrough rates at GE signals in RSV infected MRC-5 cells.**

| GE-GS boundary | Percent readthrough at 6hpi | Percent readthrough at 16hpi | Percent readthrough at 24hpi |
|---|---|---|---|
| NS1-NS2 | 21.6 | 12.9 | 11.9 |
| NS2-N | 2.3 | 3.2 | 2 |
| N-P | 3.2 | 3.5 | 4 |
| P-M | 1.5 | 2.5 | 2.3 |
| M-SH | 11.2 | 8 | 8 |
| SH-G | 0.5 | 0.6 | 0.5 |
| G-F | 0.4 | 2.6 | 2.1 |
| F-M2 | 10.5 | 9 | 8.6 |

This table shows the percentage of apparent readthrough events in MRC-5 cells within each intergenic region as indicated and listed as the region between each GE and GS signal. This is calculated from the maximum depth of read of the 5′ most gene and the read depth of the intergenic region.

**Table 4. Abundance of polycistronic messages in MRC-5 cells at 24 hours post infection.**

| | NS1 | NS2 | N | P | M | SH | G | F | M2 |
|---|---|---|---|---|---|---|---|---|---|
| First gene only | 16,863 | 10,324 | 8,310 | 9,923 | 7,960 | 4,112 | 3,159 | 3,577 | 4,165 |
| First, Second gene only (%) | 1,997 (11.8%) | 212 (2.1%) | 278 (3.3%) | 169 (1.7%) | 537 (6.7%) | 14 (0.3%) | 42 (1.3%) | 271 (7.6%) | 0 |
| First, Second, third gene only (%) | 30 (0.2%) | 1 (0.01%) | 5 (0.06%) | 10 (0.1%) | 1 (0.01%) | 0 | 1 (0.03%) | 0 | 0 |
| First, second, third, fourth gene only (%) | 3 (0.02%) | 0 | 1 (0.01%) | 0 | 0 | 0 | 0 | 0 | 0 |

This table shows the number of MRC-5 infected cell transcripts observed at 24 hours with either one gene (monocistronic) or with increasing numbers of whole additional genes present on individual mRNAs (polycistronic) after the indicated 5′ most gene. For each combination, the number of transcripts listed refers to those with the indicated structure.

from the list of characterised transcripts in S3 Table by counting the number of transcripts that contain one or more genes relative to the number of transcripts that code for just the first gene under consideration. For example, there are 10,324 transcripts containing the *NS2* gene only, compared to 212 *NS2-N* transcripts.

## L transcripts terminating at the M2 GE

The *M2-L* gene junction is unusual (see Fig 1A), with the GS signal for the *L* transcript lying upstream of the *M2* transcript GE signal. Thus, to generate full-length *L* mRNA, the polymerase must ignore the *M2* GE signal. We examined how frequently the polymerase that initiates at the *L* gene start terminates at the *M2* GE versus the *L* GE and observed transcripts that start at nucleotide 8492 (which in this MRC-5 dataset is the start location for authentic L transcripts) but terminate at 8559 (the location of transcription termination for the *M2* transcripts in this dataset). However, at both 16 and 24 hpi there was only a single transcript in each dataset with this structure and with a poly A tail greater than 20 nt in length. In contrast, there were 5 and 6 full length *L* transcripts at 16 and 24 hours respectively. Since we only consider full-length transcripts, this analysis will most likely under-represent the true number of L transcripts. For example, the read depth at 24 hpi near the 3' end of the L gene suggests that there may be as many as 240 L transcripts at this time point instead of the 6 we report based on full length reads.

## Poly A tail length in MRC-5 cells

The dRNAseq approach captures data on the length of the poly A tail and Fig 4 shows violin plots of the poly A tail length for all the transcripts used to quantitate gene expression (i.e. poly A tail > 20 nt) whose 3' ends map to the dominant GE signals for each gene. For viral mRNA the interquartile range of the poly A tail length is between 50 and 200 nt. In addition, we note that for all transcripts grouped by GE signal, between 16 and 24 hours the median poly A tail length declines (Fig 4; S1–S3 Tables).

Transcripts with short poly A tails are associated with both inefficient translation and low stability [36]. We used an in-house script to identify RSV transcripts from each time point in MRC-5 cells which had short poly A tail lengths (< = 20 nt). When these transcripts were mapped to the RSV genome, we noted that for the most part the distribution of read depth was similar except for *N*, *SH* and *G* transcripts which were notably underrepresented at 16 and 24 hpi (Fig 5). To quantify this further we processed the mapped reads with a poly A tail less than 20 nt using our ORF centric pipeline. The data at 6 hpi is harder to interpret with confidence due to the low overall read depth. Table 5 provides a more quantitative analysis of the data at 24 hpi illustrating the dramatic fall in the relative contributions of *N*, *SH* and *G* transcripts with poly A tails less than or equal to 20 nt in length.

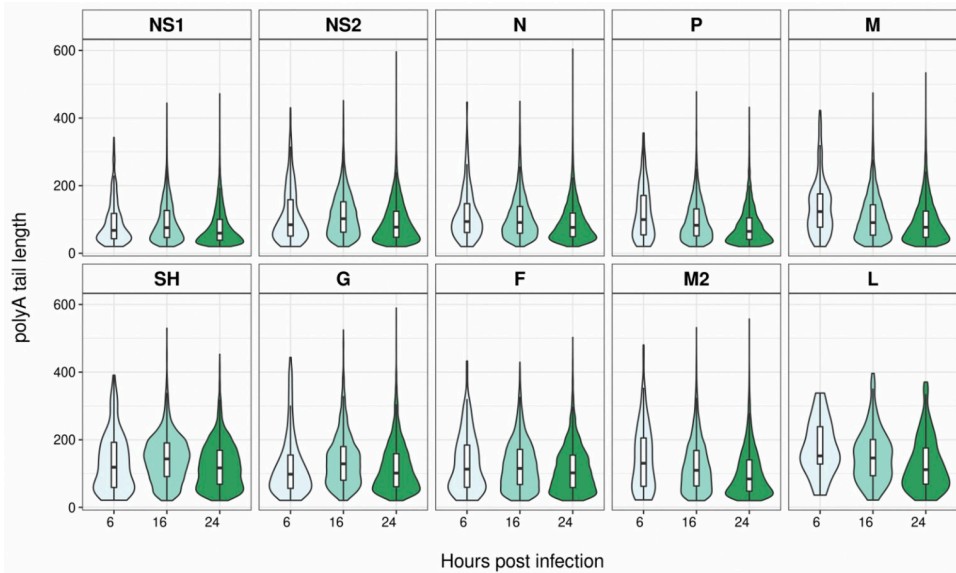

**Fig 4. Violin plots of poly A tail lengths of MRC-5 infected cell viral transcripts terminating at different GE sites.**
A series of violin plots showing the measured poly A tail lengths of transcripts that share the GE signal following each individual ORF on the RSV genome. The different time points are indicated in shades of green from light to dark for increasing hpi (6, 16 and 24). In addition, a boxplot is superimposed on each violin plot, showing the median and interquartile range.

## Variant transcripts in MRC-5 cells

As we have shown in other viral systems there are a large number of variant transcripts reported by our pipeline (S1–S3 Tables). In most cases, these likely arise from mRNA molecules that are incomplete due to nucleases or mechanical shearing. In some cases, there are micro deletions, which could potentially be an artefact of the sequencing technology or of the mapping algorithms. For example, the most frequently reported minor deletion at 24 hpi is at nucleotides 849 to 862 and there is an A-rich region at this location. Nanopore sequencing is recognised to have difficulties in accurately reporting homopolymeric runs [37]. However, there are additional transcripts that appear to result from more substantive skipping events.

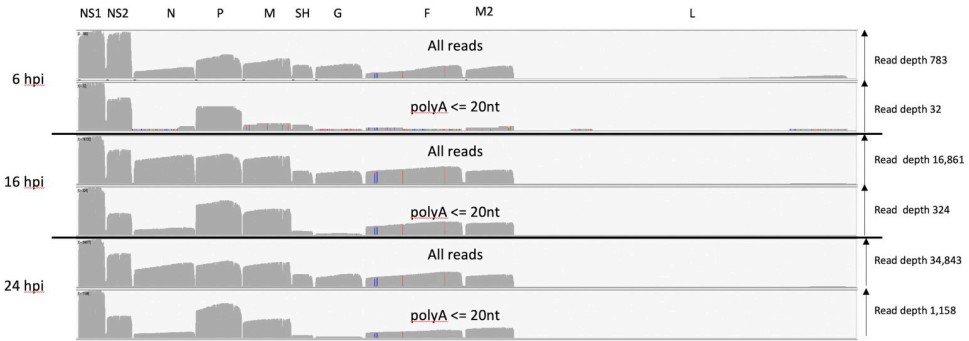

**Fig 5. MRC-5 infected cell transcripts mapped to the RSV genome and grouped by poly A tail length.** An IGV generated image of the dRNAseq reads mapped to the viral genome split into three sections representing the mRNA sequenced at 6, 16 and 24 hpi, the location of the viral ORFs is indicated along the top. Each section has two panels, the top panel shows the overall depth of reads along the genome for all the mapped reads and the lower one for only reads with a poly A length of between 1 and 20 nucleotides as reported by nanopolish software.

**Table 5. Contribution to the RSV transcriptome of transcripts with short poly A tails at 24 hpi in MRC-5 cells.**

| Feature | Gene order on viral genome | poly A < = 20 raw count | poly A < = 20 percent of total (percent of NS1) | poly A > 20 raw count | poly A > 20 percent of total (percent of NS1) |
|---|---|---|---|---|---|
| Not identified | N/A | 607 | 16.94 | 19,587 | 19.82 |
| NS1 | 1 | 1028 | 28.63 | 18,926 | 19.16 |
| NS2 | 2 | 369 | 10.30 (36) | 10,539 | 10.67 (56) |
| N | 3 | 93 | 2.59 (9) | 8,604 | 8.71 (45) |
| P | 4 | 628 | 17.52 (61) | 10,111 | 10.23 (53) |
| M | 5 | 344 | 9.60 (33) | 8,630 | 8.73 (45) |
| SH | 6 | 36 | 1.00 (4) | 4,126 | 4.18 (22) |
| G (initiation at 2nd AUG) | 7 | 26 | 0.73 (3) | 3,203 | 3.24 (17) |
| F | 8 | 101 | 2.82 (10) | 3,850 | 3.90 (20) |
| M2-1 | 9 | 186 | 5.20 (18) | 4,329 | 4.38 (23) |
| L | 10 | 0 | 0 | 6 | 0.01 (0) |
| Total of all features | | 3,584 | | 98,801 | |

This table illustrates the number of reads mapped that contain the indicated RSV ORFs as the 5′ most ORF. For each ORF raw counts and percent contribution have been calculated for transcripts with a poly A tail over 20nt and those with a poly A tail equal to or less than 20nt.

For example, screening for mRNA transcript groups where at least 10 mRNA molecules were observed reveals over 134 different transcripts with small deletions between 9 and 40 nucleotides long (S3 Table, "Deletion events" spreadsheet). This totals some 4,000 individual polyadenylated mRNA molecules with deletions of at least 9 nucleotides. This includes, for example, a transcript with a 25 nt deletion within the *NS2* gene between nucleotides 849 and 874 on the virus genome, leading to an out of frame truncation of the *NS2* ORF. For this transcript there were 168 distinct mRNA molecules with an average poly A tail length of 95 nucleotides. This same 25 nt deletion was observed in 77 sequenced transcripts at 16 hpi (S2 Table) with an average poly A tail length of 107 nt. In addition, we also see cases where there are apparently well utilised poly A sites at significant distance from classical GE signals. For example, there are some 459 transcripts that apparently have a poly A tract that begins near or on nucleotide 3090, almost 200 nt away from the dominant poly A site for the *P* gene transcripts at nucleotide 3244. This group of aberrantly polyadenylated transcripts have the potential to encode full length P protein and have an average poly A tail length of 124 nt.

There are also individual transcripts with unusual structures. For example, within the 24 hr post infection dataset there are 59 RSV transcripts with an insertion of over 100 nt (S3 Table). Examining the top 10 insertions reveals a mixture of RSV sequences, unknown sequences, and apparently one region of a human mRNA. In addition, we mapped the transcripts to concatenated RSV genomes to see if any transcripts mapped across two duplicate genomes. We identified just 89 transcripts that were each unique but did indeed apparently map across two genomes concatenated together in the same sense (S1 and S2 Files).

## RSV transcriptome in Calu-3 cells

There is the formal possibility that our MRC-5 transcription data is cell type specific and so we sequenced RSV infected Calu-3 lung epithelial cells sampled at 48hpi. Reassuringly, the alignment of RSV reads shows a broad agreement with our MRC-5 data (Fig 6). The average poly A tail length for RSV mRNA transcripts was between 90 and 150nt long (S4 Table). However, analysing the reads with a short poly A tail shows a pattern of under representation that is similar to the MRC-5 derived data but not identical (Fig 5). What is consistent with the MRC-5

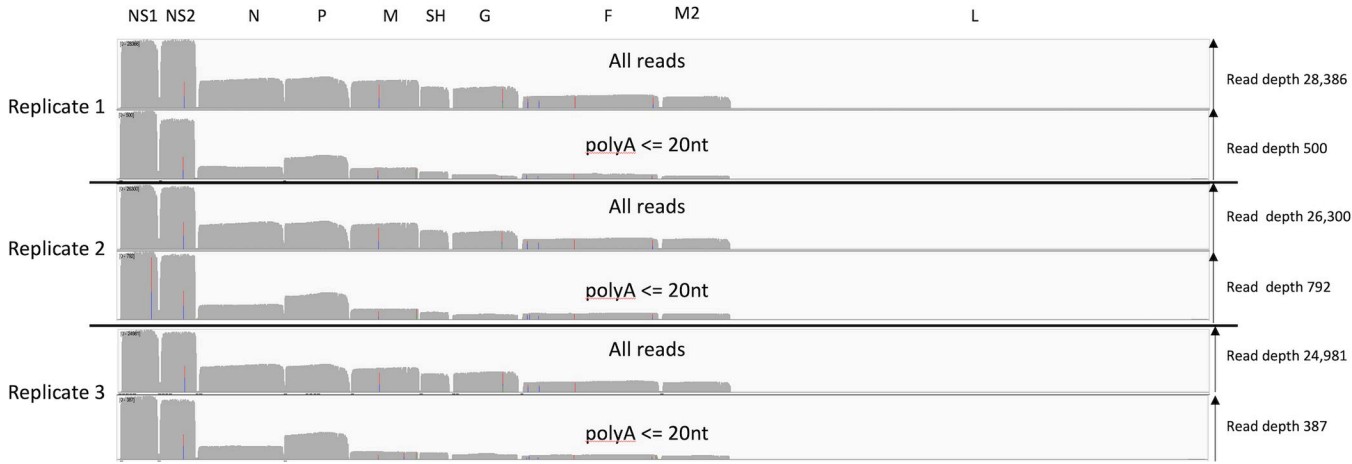

**Fig 6. Calu-3 infected cell transcripts mapped to the RSV genome and grouped by poly A tail length.** An IGV generated image of the dRNAseq reads mapped to the viral genome split into three sections representing the mRNA sequenced from Calu-3 infected cell replicates 1, 2 and 3, the location of the viral ORFs is indicated along the top. Each section has two panels, the top panel shows the overall depth of reads along the genome for all the mapped reads and the lower one for only reads with a poly A length of between 1 and 20 nucleotides as reported by nanopolish software.

data is that N, SH and G transcripts with short (<20 nt) poly A tails are, again, underrepresented (Fig 6 and S4 Table). Next, we analysed the relative contribution of RSV genes to the viral transcriptome (Table 6).

In Calu-3 cells there is notably more NS2 mRNA than in MRC-5 cells (comparing Tables 2 and 6) but as with the MRC-5 data we observe that genes can be grouped based on relative abundance. Group 1 comprises NS1 and NS2 (approximately 20%) followed by N, P and M in group 2 (approximately 9%), SH and G in group 3 (approximately 6%) followed by F and M2 (approximately 3–4%) in group 4 and L forming group 5. In the Calu-3 dataset we observed

**Table 6. List of genes from RSV and their percent contribution to the RSV derived transcriptome in Calu-3 cells at a single timepoint.**

| Feature | Gene order on viral genome | Rep 1 raw count | Rep 1 percent of total (percent of NS1) | Rep 2 raw count | Rep 2 percent of total (percent of NS1) | Rep 3 raw count | Rep 3 percent of total (percent of NS1) |
|---|---|---|---|---|---|---|---|
| **Not identified** | N/A | 5548 | 6.04 | 5542 | 7.09 | 5031 | 6.47 |
| **NS1** | 1 | 21523 | 23.45 | 18363 | 23.48 | 18354 | 23.60 |
| **NS2** | 2 | 17765 | 19.35 (83) | 14929 | 19.09 (81) | 14681 | 18.88 (80) |
| **N** | 3 | 7910 | 8.62 (37) | 6497 | 8.31 (35) | 7076 | 9.10 (39) |
| **P** | 4 | 8710 | 9.49 (41) | 7052 | 9.02 (38) | 7234 | 9.30 (39) |
| **M** | 5 | 8470 | 9.23 (39) | 7248 | 9.27 (39) | 6867 | 8.83 (37) |
| **SH** | 6 | 5809 | 6.33 (27) | 4852 | 6.21 (26) | 4782 | 6.15 (26) |
| **G (initiation at 2nd AUG)** | 7 | 5530 | 6.02 (26) | 4089 | 5.23 (22) | 5031 | 6.47 (27) |
| **F** | 8 | 3802 | 4.14 (18) | 2929 | 3.75 (16) | 2997 | 3.85 (16) |
| **M2** | 9 | 3191 | 3.48 (15) | 2742 | 3.51 (15) | 2665 | 3.43 (15) |
| **L** | 10 | 33 | 0.04 (0) | 12 | 0.02 (0) | 29 | 0.04 (0) |
| **Total of all features** | | 91789 | | 78194 | | 77760 | |

This table illustrates the number of reads mapped that contain the indicated RSV genes as the 5′ most gene. For each replicate (rep 1, 2 and 3) the raw number of individual transcripts is indicated alongside their percent contribution of those transcripts in the biological replicate. In addition, we have calculated the percent abundance of each transcript relative to NS1 abundance. Note that this table shows transcripts with a poly A tail over 20nt long

very few variant transcripts with internal deletions but we noted substantial occurrences of apparent aberrant polyA site usage (S5 Table) including, for example, some 306 transcripts with a poly adenylation site at nt 3068, some 160nt away from the dominant P gene poly A site in Calu-3 cells at nt 3228. Analysis of polycistronic mRNA production reveals a pattern of maximum rates of readthrough at the NS1-NS2 boundary, modest readthrough at M-SH and F-M2 boundaries and low readthrough at the other boundaries (S6 Table). This reflects the observations in MRC-5 cells (Table 4). Overall we note that the pattern and abundance of transcripts produced are similar across the three replicates (S5 Table). We did not observe any transcripts corresponding to the M2 GS–L GE transcript found at low abundance in MRC-5 cells (S5 Table).

## Discussion

Our dRNAseq dataset provides a detailed analysis of RSV mRNA expression at three time-points in MRC-5 cells, a genetically normal lung cell line and subsequently at a single time-point in triplicate in Calu-3 cells (a human lung adenocarcinoma line). By using dRNAseq we can quantify viral mRNA without the potentially confounding artefacts from, for example, reverse transcription or PCR amplification. In addition, because our analysis is limited to transcripts for which we have quantitated the poly A tail length we can also be confident that we are observing and quantitating only mRNA, and not genomic/antigenome RNA or immature mRNA. Examining the RSV transcriptome in two infected lung epithelial cells suggests that our observations will be applicable to other human cell types. Clearly, our platform can be translated to study other RSV strains and infected primary cells or clinical samples to provide an even broader picture of RSV transcription.

Our data shows a stepped gradient of individual transcripts decreasing in abundance from the 3′ to the 5′ end of the genome. However, there is minimal evidence for an attenuation at some gene junctions, rather there are groups of genes with similar transcript levels. In MRC-5 cells at 16 and 24 hpi we see four groups (NS1, followed by NS2/N/P/M, followed by SH/G/F/M2 followed by L) whereas in Calu-3 cells there is a similar but not identical step down of abundance in 5 transcript groups (NS1/NS2 followed by N/P/M then SH/G then F/M2 and finally L). In contrast to previous recent reports using qRT-PCR [26] or our own prior Illumina RNAseq data [25] we do not see elevated levels of *G* protein transcript abundance in MRC-5 or Calu-3 cells. These discrepancies likely result from reverse transcription and PCR based steps that introduce unintended bias in the quantification of transcripts. Of note, our dRNAseq data is more aligned with early RSV data based on slot blots and UV inactivation studies [17, 23]. Indeed, a recent analysis of transcript abundance in related paramyxoviruses reported that for illumina based approaches at least, the techniques used to enrich for viral mRNA and sequence does affect abundance estimates. They went on to show that techniques which can distinguish the strand sense of the Illumina based sequence data had a significant impact on transcript abundance estimation [38]. This illustrates the potential for genome and antigenome RNA to confound attempts to determine mRNA abundance since they too will be enriched by RNA complementarity in any poly A based enrichment strategy. Nonetheless it is notable that for several gene pairs in both cell lines their abundance does not decline at the gene junction, and in some cases, such as *N-P* there is marginally more mRNA from the downstream gene. For the *P* and *N* transcripts this may be due to the *N* mRNA being longer (and therefore more prone to degradation) as dRNAseq is more likely to successfully read the full length of shorter mRNA molecules, and indeed the peaks for *N* and *P* sequence coverage are similar, indicating that this may well be the case. However, in the infected MRC-5 cells, there are more *F then G* transcripts since *G* transcripts are shorter than *F* transcripts this argument

does not explain these results. It is possible that *G* mRNA is less stable than *F* mRNA in this cell line, accounting for the difference in steady-state abundance of these transcripts. This highlights the likely role of mRNA stability in determining the steady state levels of mature RSV mRNA species. It is also possible that there is some feature of G mRNA, such as secondary structure, that makes it more difficult to sequence such that it is underrepresented in the read depth analysis. There is also a formal possibility that on occasion the viral polymerase ignores a GS signal but can progress to the subsequent GS and then re-initiate transcription. Evidence that this was reported for a paramyxovirus, SV41, in which *M* transcripts are exclusively dicistronic *M-F* mRNAs, but monocistronic *F* transcripts are also produced, suggesting that some polymerase can scan from the *P* GE signal to the *F* GS signal, without transcribing *M* [39].

Our data shows for the first time the relative levels of polycistronic mRNA by RSV, showing for the first time the relative levels of different polycistronic messages produced by the viral polymerase. An early study where RSV transcripts were analysed by Northern blot revealed polycistronic transcripts containing two or three genes, but their expression levels were not determined [11]. In our data, in both cell lines we find that the *SH-G* intergenic appears to have the strongest GE signal and therefore the fewest readthrough events with the *NS1* GE signal being the weakest. These findings are supported by previous work [11, 25, 40]. However, the significance of the strength of GE signals is unclear as previous work suggests that relatively high level of readthrough at the *NS1* and *NS2* GE signals does not play a role in viral replication, either *in vitro* or *in vivo* [41].

In addition to the canonical transcripts corresponding to the viral genes, we also examined the levels of the M2-L transcript which we only detected in infected MRC-5 cells. The RSV M2-L gene junction is unusual in that the L GS signal lies upstream (rather than downstream) of the M2 GE signal. It was previously shown that the RSV polymerase can access the L GS signal by scanning backwards from the M2 GE signal following M2 mRNA release [19]. However, the M2 GE signal has a similar termination efficiency as other RSV GE signals [42], and so if the polymerase simply responded to signals as it encounters them, it would be expected to release the L transcript at the M2 GE signal, primarily resulting in a 68 nt M2-L transcript (not including the poly A tail) and only producing full-length L mRNA as a readthrough transcript, similar to a polycistronic mRNA. However, our data shows that L mRNA is the major transcript produced from the L GS signal in both cell lines. If we limit our analysis to full-length transcripts, the L mRNA was produced at a 5-6-fold excess over the M2-L transcript in infected MRC-5 cells, and as noted in the results section, this is an under-representation of full-length L mRNA due to its large size. If we consider that the L mRNA 3′ end reads represent full-length L mRNAs then there was a 240-fold excess of L mRNA over M2-L transcripts at 24 hpi in MRC-5 cells. An explanation for why the polymerase usually generates a full-length L mRNA versus the M2-L transcript comes from work with vesicular stomatitis virus (VSV), which showed that the polymerase can only recognize a GE signal positioned beyond a certain distance from the GS signal [43]. This probably reflects the need for the polymerase to cap the mRNA before it can accurately recognize a GE signal [44]. We also examined M2-L transcripts with a limited (<20 nt) poly A tail length, as in another study it was shown that VSV polymerase that does not cap the RNA can recognize a GE signal but fails to properly polyadenylate the mRNA [45]. However, there was no evidence of such M2-L transcripts. Thus, we conclude that is probably insufficient distance before the M2 GE for it to be recognized by a polymerase that initiated at the L GS, allowing the polymerase to disregard this GE signal and synthesize full-length L mRNA, as suggested previously [43]. We did not detect the M2-L transcript in the infected Calu-3 cells which underlines the rarity of such transcripts.

We observed a number of unusual transcripts in both infected MRC-5 and Calu-3 datasets, some with minor deletions that could reflect the limitations of nanopore sequencing technology. However, we observed transcripts with more substantial deletions in the infected MRC-5 dataset. In one example we detected the same 25 nt deletion in *NS2* transcripts in our datasets at both 16 and 24 hpi. Given the number of distinct transcripts with the same deletion and their presence in independent samples suggests that these deletions are genuine and reasonably frequent. There is the possibility that defective interfering viruses were present in the input virus or they could be generated during the course of the infection in either cell line. However, it is worth noting that defective interfering viruses derived by internal deletions in the genome should generate transcripts reflecting such deletions. But we were only able to detect evidence of small deletions happening in our datasets that were low in numbers (S3 and S5 Tables). We cannot exclude the possibility that some rearrangements we detected could occur as an artefact of the sequencing technique, for example, at the DNA-RNA ligase step of sample preparation. However, we previously reported rare splicing events in the adenovirus transcriptome and went on to validate these transcripts (as few as 10 molecules in over 1.2 million reads) by directed RT-PCR [27]. Additionally, our analysis of the adenovirus transcriptome showed a broad range of mRNA species made amongst a small number of dominant transcripts. This concept was subsequently independently verified for a different serotype of adenovirus, again using dRNA-seq [46]. If the transcripts detected in the current study are genuine, it suggests the RSV polymerase has the potential to skip short distances when transcribing mRNA. Isolates of RSV have been found with small duplications in viral G gene, and defective interfering genomes are frequently generated, indicating that the viral polymerase can dissociate and reassociate with the template during genome replication [47–50]. It is not surprising to see a similar background of unusual mRNAs that do not match the classical transcripts in RSV infected cells. Potentially the higher abundance of transcripts with more substantial deletions in the infected MRC-5 cells compared to the Calu-3 may reflect the slightly higher MOI used for this infection.

We also noted the poly A tail lengths of the viral transcripts are similar to human transcripts [33] in both cell lines. To our knowledge this is the first time poly A tail length has been reported for RSV. Notably, these poly A tails are added by the viral polymerase but without any clear mechanism to explain how the length is determined or why they are similar to host mRNA which is polyadenylated in the nucleus. In addition, between 16 and 24 hpi in MRC-5 cells there is a noticeable decline in poly A tail length across all the different mRNA classes. We have previously noted a decline in poly A tail length during adenovirus infections whose mRNA are made by host RNA pol II [27] and even earlier reports noted a similar effect during coronavirus replication [51]. Therefore, a decline in poly A tail length may simply reflect a decline in host cell function as infection progresses. Potentially of greater significance is the observation that transcripts with very short poly A tails are consistently underrepresented for the *N*, *SH* and *G* genes in both infected MRC-5 and Calu-3 cells (Fig 5, Table 5 and S4 Table). We retrospectively analysed our previously published SARS-CoV-2 transcript data [31] and failed to observe any significant differences in the distribution of SARS-CoV-2 transcripts with very short poly A tails. That three RSV genes in two independent MRC-5 datasets (16 hpi and 24 hpi) and all three Calu-3 datasets show the same pattern of underrepresentation is notable and may reflect differences in rates of translation or mRNA stability both of which can be governed by poly A tail length [36]. In principle, the underrepresentation of *N*, *SH* and *G* may reflect higher rates of translation for those transcripts and experiments to examine this possibility are underway.

In conclusion, our dataset is the first dRNAseq transcriptomic analysis of RSV mRNAs in two independent lung epithelial cell lines. This dataset provides fine detail of the

transcriptomic repertoire of RSV in cell culture. Our findings align closely with early studies that attempted to determine the relative abundance of RSV transcripts and provides a robust quantitative dataset. We believe this analysis highlights the potential for inaccurate measurements when trying to estimate RSV mRNA loads by indirect methods such as qRT-PCR or short-read based RNAseq techniques that is often the end point used to assess vaccine or antiviral drug activities, and that this technical insight is broadly relevant to the study of viral gene transcription.

## Supporting information

**S1 Table. RSV characterised transcripts at 6 hours post infection.** The table shows the list of transcript groups identified by the pipeline. For each transcript group it is noted how many sequence reads belong to the group, the average poly A length, the features present on the transcript (as well as the predicted splice acceptor/donor pair usage if there is an apparent intron—in the case of RSV this is clearly not genuine splicing) and what ORFS are found that are present in the features file that accompanied the analysis.
(XLSX)

**S2 Table. RSV characterised transcripts at 16 hours post infection.** The first datasheet shows the list of transcript groups identified by the pipeline. For each transcript group it is noted how many sequence reads belong to the group, the average poly A length, the features present on the transcript (as well as the predicted splice acceptor/donor pair usage if there is an apparent intron—in the case of RSV this is clearly not genuine splicing) and what ORFS are found that are present in the features file that accompanied the analysis.
(XLSX)

**S3 Table. RSV characterised transcripts at 24 hours post infection.** The first datasheet shows the list of transcript groups identified by the pipeline. For each transcript group it is noted how many sequence reads belong to the group, the average poly A length, the features present on the transcript (as well as the predicted splice acceptor/donor pair usage if there is an apparent intron—in the case of RSV this is clearly not genuine splicing) and what ORFS are found that are present in the features file that accompanied the analysis.
(XLSX)

**S4 Table. RSV gene expression in Calu-3 cells.** This table shows the relative contribution of each RSV gene at 48 hours post infection in Calu-3 cells. The first sheet includes average poly A length data for all reads. In addition, the second sheet shows just relative gene expression data, but alongside the relative contribution of each gene when only transcripts with a short poly A tail (<20nt) are considered.
(XLSX)

**S5 Table. RSV characterised transcripts at 48 hours post infection, Calu-3 replicates 1, 2 and 3.** Each datasheet shows the list of transcript groups identified by the pipeline. For each transcript group it is noted how many sequence reads belong to the group, the average poly A length, the features present on the transcript (as well as the predicted splice acceptor/donor pair usage if there is an apparent intron—in the case of RSV this is clearly not genuine splicing) and what ORFS are found that are present in the features file that accompanied the analysis.
(XLSX)

**S6 Table. Analysis of readthrough events in Calu-3 cells compared to MRC-5 cells.** This table shows the number of transcripts observed at 24 hours with either one gene (monocistronic) or with increasing numbers of whole additional genes present on individual mRNAs

(polycistronic) after the indicated 5¢ most gene. For each combination, the number of transcripts cited refers to only transcripts that have that indicated structure. Data for each replicate is shown seperately and derived from analysis nof S6 Table. Also reproduced is the data from Table 4 in the manuscript which is derived from MRC5 cells.
(XLSX)

**S1 File. Zipped SAM file of reads mapping to concatenated genome.** This is a sorted SAM file of reads that mapped to the FF orientation of the RSV genome in S2 File across the two concatenated genomes.
(ZIP)

**S2 File. Zipped FASTA file of concatenated RSVA genomes.** This is a FASTA format file of the RSVA genome concatenated sequentially (i.e.e in a FF orientation) and used to examine the potential for transcripts that may have unusual rearrangements with respect to the RSV genome.
(ZIP)

## Author Contributions

**Conceptualization:** Julian Hiscox, Jane A. McKeating, Rachel Fearns, David A. Matthews.

**Data curation:** David A. Matthews.

**Formal analysis:** I'ah Donovan-Banfield, Rachel Fearns, David A. Matthews.

**Funding acquisition:** Jane A. McKeating, David A. Matthews.

**Investigation:** I'ah Donovan-Banfield, Rachel Milligan, Sophie Hall, Tianyi Gao, Eleanor Murphy, Jack Li, Ghada T. Shawli, Xiaodong Zhuang, David A. Matthews.

**Methodology:** Xiaodong Zhuang, David A. Matthews.

**Project administration:** David A. Matthews.

**Software:** David A. Matthews.

**Supervision:** Jane A. McKeating, David A. Matthews.

**Validation:** I'ah Donovan-Banfield, David A. Matthews.

**Visualization:** I'ah Donovan-Banfield, David A. Matthews.

**Writing – original draft:** I'ah Donovan-Banfield, Julian Hiscox, Rachel Fearns, David A. Matthews.

**Writing – review & editing:** I'ah Donovan-Banfield, Rachel Milligan, Sophie Hall, Tianyi Gao, Eleanor Murphy, Jack Li, Ghada T. Shawli, Julian Hiscox, Xiaodong Zhuang, Jane A. McKeating, Rachel Fearns, David A. Matthews.

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
