## [Decision Letter · Decision Letter 0]

16 May 2022

PONE-D-22-03411Direct RNA sequencing of Respiratory Syncytial Virus infected human cells generates a detailed overview of RSV polycistronic mRNA and transcript abundance.PLOS ONE

Dear Dr. Matthews,

Thank you for submitting your manuscript to PLOS ONE. After careful consideration, we feel that it has merit but does not fully meet PLOS ONE’s publication criteria as it currently stands. Therefore, we invite you to submit a revised version of the manuscript that addresses the points raised during the review process.

We look forward to receiving your revised manuscript.

Kind regards,

Paulo Lee Ho, Ph.D.

Academic Editor

PLOS ONE

Journal Requirements:

Reviewers' comments:

Reviewer's Responses to Questions

**Comments to the Author**

1. Is the manuscript technically sound, and do the data support the conclusions?

Reviewer #1: Yes

Reviewer #2: Partly

2. Has the statistical analysis been performed appropriately and rigorously? 

Reviewer #1: N/A

Reviewer #2: N/A

3. Have the authors made all data underlying the findings in their manuscript fully available?

Reviewer #1: Yes

Reviewer #2: Yes

4. Is the manuscript presented in an intelligible fashion and written in standard English?

Reviewer #1: Yes

Reviewer #2: Yes

5. Review Comments to the Author

Reviewer #1: The authors used the ONT direct RNA sequencing (dRNAseq) method to examine the RSV transcriptome at three time points. The quantitative and qualitative measures of the sequenced RSV transcripts were included in their study. The quantitative measures comprised the abundance of expression of the various genes at different time points. The qualitative measures included cistrons characterization, PolyA tail length, and readthrough and intergenic structures. The paper is nicely written, and the results are presented in a logical and understandable manner. The interpretations are correct, and prior publications back them up. The findings of this investigation provide support for the final conclusions.

I have a few minor comments:

1) The section where they mapped the dRNAseq reads to the human genome looks to be a last-minute addition. Because of the limited depth of coverage of human dRNAseq reads discovered in the samples, the authors did not say anything about this section. I propose that the authors remove this part or offer additional analysis or data to back it up.

2) For consistency, the authors should use either “PolyA” or “Poly A”. I have seen both forms in the manuscript.

3) For clarity, I suggest spelling out the word nucleotide, instead of “nt” when refereeing to specific nucleotide positions. In this context, nucleotide is not used as a unit.

4) When talking about the variant transcripts, instead of saying “less than 100 transcripts, it is better to say the exact number.

5) it was mentioned many times that in-house scripts were used. These scripts are not deposited on Zenodo with the data. I believe they should be deposited.

Reviewer #2: Respiratory Syncytial Virus (RSV) is a major worldwide respiratory pathogen responsible for severe respiratory infections in infants and young children. In this study, the authors used the Oxford Nanopore device to perform direct RNA sequencing of viral transcripts isolated from human cells infected with RSV. Then, the authors characterized the viral mRNAs including polycistronic mRNAs, determined the rates of readthrough at gene end signal sequences and the length of transcripts’ polyA tails. The manuscript is well written and nicely describe the complexity of the RSV gene expression. However, the main concern is that the data are derived from a single experiment using a single cell line with no replicate for the three investigated time points. It is thus impossible to know how reproducible these data are.

Specific comments:

1- Introduction: RSV genome contains 10 genes encoding for 11 proteins. The authors did not mention M2-2.

2- Materials and Methods: How was generated the RSV A2 stock used in this study? A high MOI (10 pfu/cell) was used. Thus, it is possible that the authors infected the MRC-5 cells with a stock of RSV A2 that already contained defective interfering (DI) particles that could have affected the data. Does RSV A2 replicate efficiently in MRC-5 cells? To my knowledge this cell line is not widely used in the RSV field.

3- Results: first paragraph “MOI of 10 pfu/ml” should be “10/pfu/cell”.

4- The authors indicated that they have detected “molecules with apparent short deletions within the body of the transcript’. As the authors mentioned this could have been generated by the viral polymerase during the infection. However, as described above, we cannot exclude that the transcripts that contain deletions could have been derived from DI particles that were already present in the original inoculum.

5- Table 2: Did the authors detect any M2-2 transcripts?

6- L transcripts which terminate at the M2 GE paragraph: I did not understand the sentence “At both 16 and 24 hours post infection there was one transcript…..”.

6. PLOS authors have the option to publish the peer review history of their article (what does this mean?). If published, this will include your full peer review and any attached files.

Reviewer #1: No

Reviewer #2: No

---

## [Author Response · Author response to Decision Letter 0]

8 Jul 2022

Reviewer #1: The authors used the ONT direct RNA sequencing (dRNAseq) method to examine the RSV transcriptome at three time points. The quantitative and qualitative measures of the sequenced RSV transcripts were included in their study. The quantitative measures comprised the abundance of expression of the various genes at different time points. The qualitative measures included cistrons characterization, PolyA tail length, and readthrough and intergenic structures. The paper is nicely written, and the results are presented in a logical and understandable manner. The interpretations are correct, and prior publications back them up. The findings of this investigation provide support for the final conclusions.

I have a few minor comments:

1) The section where they mapped the dRNAseq reads to the human genome looks to be a last-minute addition. Because of the limited depth of coverage of human dRNAseq reads discovered in the samples, the authors did not say anything about this section. I propose that the authors remove this part or offer additional analysis or data to back it up.

Response: We have removed this section and the table at the reviewer’s request. Section deleted on page 14, line 47 (results section) and page 17, line 40 (Discussion).

2) For consistency, the authors should use either “PolyA” or “Poly A”. I have seen both forms in the manuscript.

Response: Corrected to “Poly A” throughout as requested.

3) For clarity, I suggest spelling out the word nucleotide, instead of “nt” when refereeing to specific nucleotide positions. In this context, nucleotide is not used as a unit.

Response: Corrected as requested throughout.

4) When talking about the variant transcripts, instead of saying “less than 100 transcripts, it is better to say the exact number.

Response: Corrected as requested (Page 14, line 43).

5) it was mentioned many times that in-house scripts were used. These scripts are not deposited on Zenodo with the data. I believe they should be deposited.

Response: We have now deposited the scripts used to classify transcripts according to poly A usage and length on Zenodo with instructions and included this information in the materials and methods section (page 7, lines 25-29). The only other scripts used are from a prior publication and are available as already described in that manuscript and from Zenodo as outlined in the materials and methods.

Reviewer #2: Respiratory Syncytial Virus (RSV) is a major worldwide respiratory pathogen responsible for severe respiratory infections in infants and young children. In this study, the authors used the Oxford Nanopore device to perform direct RNA sequencing of viral transcripts isolated from human cells infected with RSV. Then, the authors characterized the viral mRNAs including polycistronic mRNAs, determined the rates of readthrough at gene end signal sequences and the length of transcripts’ polyA tails. The manuscript is well written and nicely describe the complexity of the RSV gene expression. However, the main concern is that the data are derived from a single experiment using a single cell line with no replicate for the three investigated time points. It is thus impossible to know how reproducible these data are.

Response: This is a reasonable point to make about the data and we do not wish to make claims broader than our data supports. However, in terms of reproducibility of the sequence analysis, we note that we examined three time points with independent extractions and sequencing runs and the data for the 12 and 24 h time points are almost identical. Moreover, the data at 6 hours is similar enough to support the key claims we make given how early post infection that sample was taken. In addition, we have amended the manuscript to make it very clear that further direct RNA sequencing in a wider range of cell lines and experimental setups will be valuable. Specifically, we have inserted this paragraph in the discussion on page 15, lines 9-12: “However, further work is required to determine if this pattern is applicable to other human cell lines or strains of human RSV. Moreover, although the three data sets are independent of each other, additional time points and repeats would broaden further the applicability of our analysis.”

Specific comments:

1- Introduction: RSV genome contains 10 genes encoding for 11 proteins. The authors did not mention M2-2.

Response: Corrected as requested by inserting the following in the introduction on page 3, lines 13-16: “The 10 genes code for corresponding proteins except M2 which codes for both M2-1 (the 5’ most open reading frame) and M2-2 which is expressed by an unusual ribosome shunting mechanism and does not have a separate transcript(13).”

2- Materials and Methods: How was generated the RSV A2 stock used in this study? A high MOI (10 pfu/cell) was used. Thus, it is possible that the authors infected the MRC-5 cells with a stock of RSV A2 that already contained defective interfering (DI) particles that could have affected the data. Does RSV A2 replicate efficiently in MRC-5 cells? To my knowledge this cell line is not widely used in the RSV field.

Response: We apologize that the stated MOI was incorrectly written as 10 pfu/ cell when in fact it was 3 TCID50/cell and we have now corrected this error. The purpose of using this MOI was to ensure most of the cells were infected thus increasing the number of virus derived transcripts that we would subsequently sequence. We have also added a short description of how the stocks were grown and evaluated for infectivity (Methods section page 6, lines 7-10). They were grown and assayed by TCID50 in HEp2 cells which is standard in the field as this cell line does generate stocks with the highest titre. It is also worth noting (as we further discuss below in response to point 4) that at the least, significant contamination by a DI virus with internal deletions would have been readily detected by direct RNA sequencing as it would have resulted in large numbers of viral mRNA with large deletions/fusions between disparate parts of the genome. 

MRC5 cells were selected as they are derived from a human lung and are genetically normal (diploid with 46, XY karyotype). We are not aware of any concerns that MRC-5 cells are not suitable for RSV and indeed this cell line is recommended for isolation of RSV from clinical samples (Enhanced isolation of respiratory syncytial virus in cell culture. J Clin Microbiol. 1986 Apr; 23(4): 800–802). Other widely used cell lines are arguably less relevant. For example, HEp-2 cells have been found to be a HeLa cell contaminated line and while A549 cells, for example, are lung cells they are cancerous (lacking p14ARF) and hypotriploid. 

3- Results: first paragraph “MOI of 10 pfu/ml” should be “10/pfu/cell”.

Response: Corrected as requested (page 8 line 5) and to reflect the use of TCID50 assay method.

4- The authors indicated that they have detected “molecules with apparent short deletions within the body of the transcript’. As the authors mentioned this could have been generated by the viral polymerase during the infection. However, as described above, we cannot exclude that the transcripts that contain deletions could have been derived from DI particles that were already present in the original inoculum.

Response: We have included a short section in the discussion (page 16 line 46, continued into page 17 line 5) to alert the reader to the possibility that there may be DI particles present in the inoculum as follows: “There is also the formal possibility of defective interfering viruses being present in the input virus or being generated during the course of the infection. However, it is worth noting that defective interfering viruses derived by internal deletions in the genome should generate transcripts reflecting such deletions. But we were only able to detect evidence of small deletions happening in our datasets that were overall low in numbers (S3 Table)”.

5- Table 2: Did the authors detect any M2-2 transcripts?

Response: It is well established that M2-2 is expressed from the same mRNA as M2-1 by a ribosomal shunt mechanism. As mentioned earlier we have amended the introduction to make this clear. However, if the referee is asking if we detected any novel transcripts that could code for M2-2 as a distinct mRNA, the answer is that this is not reliably possible with this technique as it sequences the molecules poly A tail first and cannot detect the 5’ cap of mRNA. To clarify this to the reader we have added a short statement to the results section (page 9 lines 22-26) explaining that confident identification of low level usage of any novel GS signal would be problematic: “Allied to this observation, whilst we observe patterns of transcript mapping that aligns closely with the expected GS signals, it would be difficult using this approach to identify low level use of any novel GS signals as the technique is insensitive to the 5’ cap structure and thus unable to reliably identify full length transcripts.”

6- L transcripts which terminate at the M2 GE paragraph: I did not understand the sentence “At both 16 and 24 hours post infection there was one transcript…..”.

Response: We have revised our explanation which hopefully clears up any misunderstanding. The new text (page 12 lines 17 - 20) reads as follows: “However, at both 16 and 24 hours post infection there was just one such transcript in each dataset with this structure and with a poly A tail greater than 20 nt in length. In the same datasets, by comparison, there were 5 and 6 full length L transcripts at 16 and 24 hours respectively.”

---

## [Decision Letter · Decision Letter 1]

16 Aug 2022

PONE-D-22-03411R1Direct RNA sequencing of Respiratory Syncytial Virus infected human cells generates a detailed overview of RSV polycistronic mRNA and transcript abundance.PLOS ONE

Dear Dr. Matthews,

Thank you for submitting your manuscript to PLOS ONE. After careful consideration, we feel that it has merit but does not fully meet PLOS ONE’s publication criteria as it currently stands. Therefore, we invite you to submit a revised version of the manuscript that addresses the points raised during the review process.

1) The major concern is related to the results that are based on a single experiment and a single cell line. In view of this, the authors should consider to evaluate the extension of the findings in other cell lines or confirmation by additional experiments;2) Please, see the comments raised by both reviewers.

We look forward to receiving your revised manuscript.

Kind regards,

Paulo Lee Ho, Ph.D.

Academic Editor

PLOS ONE

Reviewers' comments:

Reviewer's Responses to Questions

**Comments to the Author**

1. If the authors have adequately addressed your comments raised in a previous round of review and you feel that this manuscript is now acceptable for publication, you may indicate that here to bypass the “Comments to the Author” section, enter your conflict of interest statement in the “Confidential to Editor” section, and submit your "Accept" recommendation.

Reviewer #1: All comments have been addressed

Reviewer #2: (No Response)

2. Is the manuscript technically sound, and do the data support the conclusions?

Reviewer #1: Yes

Reviewer #2: Partly

3. Has the statistical analysis been performed appropriately and rigorously? 

Reviewer #1: Yes

Reviewer #2: Yes

4. Have the authors made all data underlying the findings in their manuscript fully available?

Reviewer #1: Yes

Reviewer #2: Yes

5. Is the manuscript presented in an intelligible fashion and written in standard English?

Reviewer #1: Yes

Reviewer #2: Yes

6. Review Comments to the Author

Reviewer #1: I reviewed all the changes that you introduced in response to my initial criticism and I have no further comments.Thank you for addressing them all.

Reviewer #2: The authors modified the discussion to indicate that their study is based on a single experiment and a single cell line. However, in my opinion, it is insufficient to publish a manuscript that relies on a single experiment. I think that the authors should at the very least repeat their experiment once.

I apologize for the misunderstanding regarding Table 2. All ORFs but M2-2 are mentioned in this table. The authors should replace M2-1 by M2 and ORFs by genes in the title or indicate if they found reads corresponding to the M2-2 sequence.

7. PLOS authors have the option to publish the peer review history of their article (what does this mean?). If published, this will include your full peer review and any attached files.

Reviewer #1: No

Reviewer #2: No

---

## [Author Response · Author response to Decision Letter 1]

21 Sep 2022

Review Comments to the Author

Reviewer #1: I reviewed all the changes that you introduced in response to my initial criticism and I have no further comments.Thank you for addressing them all.

Reviewer #2: The authors modified the discussion to indicate that their study is based on a single experiment and a single cell line. However, in my opinion, it is insufficient to publish a manuscript that relies on a single experiment. I think that the authors should at the very least repeat their experiment once.

I apologize for the misunderstanding regarding Table 2. All ORFs but M2-2 are mentioned in this table. The authors should replace M2-1 by M2 and ORFs by genes in the title or indicate if they found reads corresponding to the M2-2 sequence.

Response: We are pleased that reviewer 1 is happy with the manuscript after our previous resubmission. 

Reviwer 2 had two points, we would answer the second request first by stating that we have altered the titles and wording as requested. We are unable to confidently assign any reads to M2-2 as these could arise as a result of 5’ degradation of a bone fide M2 gene mRNA giving the false impression that this was an M2-2 only transcript. Unfortunately, nanopore sequencing does not allow for confident detection of the 5’ cap at this time. We did address this issue in our first rebuttal and added text in the manuscript at that time which is still present in this version (page 9 lines 22-26).

For the first and main point about reproducibility, as we discussed with yourselves, we included a new dataset derived from three independent RSV infections of Calu-3 cells harvested at 48 hours post infection. This RNA was harvested by a team at Oxford using a different preparation of RSV strain A2 as part of ongoing research at Oxford which we are collaborating on. As you may imagine we needed to re-write some sections of the manuscript, re-edit for clarity and add new figures, tables and supplementary tables as well as new links to raw data. The main additions and changes are:

1. New figure 6 detailing the distribution of mapped reads on the RSV genome alongside mapped reads with a poly A tail of less than 20nt (e.g. similar to figure 5).

2. A new table, Table 6, detailing the contribution of different genes to overall gene abundance in each of the three replicates (e.g. similar to Table 2)

3. A new section in the results section (pages 15 and 16) describing our findings.

4. Modified discussion integrating our new findings with our previous observations in MRC-5 cells.

5. Additional text in the methods section (p6) outlining how the Calu-3 infections were performed.

6. Modified links in the data availability and code availability sections to easily allow the reader to access the raw fastq files and the software needed for the ORF-centric analysis described.

7. Three new supplementary tables (S4, S5 and S6) for the Calu-3 infected cell data covering RSV gene expression (S4), characterisation of transcripts (S5) and polycistronic gene analysis (S6). These tables are described on p26.

---

## [Decision Letter · Decision Letter 2]

12 Oct 2022

Direct RNA sequencing of Respiratory Syncytial Virus infected human cells generates a detailed overview of RSV polycistronic mRNA and transcript abundance.

PONE-D-22-03411R2

Dear Dr. Matthews,

We’re pleased to inform you that your manuscript has been judged scientifically suitable for publication and will be formally accepted for publication once it meets all outstanding technical requirements.

Kind regards,

Paulo Lee Ho, Ph.D.

Academic Editor

PLOS ONE

Additional Editor Comments (optional):

Reviewers' comments:

Reviewer's Responses to Questions

**Comments to the Author**

1. If the authors have adequately addressed your comments raised in a previous round of review and you feel that this manuscript is now acceptable for publication, you may indicate that here to bypass the “Comments to the Author” section, enter your conflict of interest statement in the “Confidential to Editor” section, and submit your "Accept" recommendation.

Reviewer #2: All comments have been addressed

2. Is the manuscript technically sound, and do the data support the conclusions?

Reviewer #2: Yes

3. Has the statistical analysis been performed appropriately and rigorously? 

Reviewer #2: Yes

4. Have the authors made all data underlying the findings in their manuscript fully available?

Reviewer #2: Yes

5. Is the manuscript presented in an intelligible fashion and written in standard English?

Reviewer #2: Yes

6. Review Comments to the Author

Reviewer #2: (No Response)

7. PLOS authors have the option to publish the peer review history of their article (what does this mean?). If published, this will include your full peer review and any attached files.

Reviewer #2: No

---

## [Editor Report · Acceptance letter]

1 Nov 2022

PONE-D-22-03411R2 

Direct RNA sequencing of Respiratory Syncytial Virus infected human cells generates a detailed overview of RSV polycistronic mRNA and transcript abundance. 

Dear Dr. Matthews:

I'm pleased to inform you that your manuscript has been deemed suitable for publication in PLOS ONE. Congratulations! Your manuscript is now with our production department. 

Kind regards, 

on behalf of

Dr. Paulo Lee Ho 

Academic Editor

PLOS ONE